# Creating bulk ultrastable glasses by random particle bonding

Misaki Ozawa[1], Yasutaka Iwashita[2], Walter Kob [3] & Francesco Zamponi [1] ✉

A recent breakthrough in glass science has been the synthesis of ultrastable glasses via physical vapor deposition techniques. These samples display enhanced thermodynamic, kinetic and mechanical stability, with important implications for fundamental science and technological applications. However, the vapor deposition technique is limited to atomic, polymer and organic glass-formers and is only able to produce thin film samples. Here, we propose a novel approach to generate ultrastable glassy configurations in the bulk, via random particle bonding, and using computer simulations we show that this method does indeed allow for the production of ultrastable glasses. Our technique is in principle applicable to any molecular or soft matter system, such as colloidal particles with tunable bonding interactions, thus opening the way to the design of a large class of ultrastable glasses.

Glasses are usually produced by slowly cooling a melt, with slower cooling leading to samples of higher stability[1]. Accessing highly stable glasses is important not only to answer fundamental questions, such as the existence of a phase transition to an ideal glass at the Kauzmann temperature[2–4], but also for technological applications, such as creating glasses with exceptionally high strength and hardness[5,6] or with strongly reduced energy dissipation[7]. However, in practice, the range of accessible cooling rates is quite limited both in experiments and (even more) in numerical simulations, and hence one does not have much leeway to modify the properties of the glass since these depend on a logarithmic manner on the quench rate[2]. About a decade ago, a breakthrough was achieved by Ediger and coworkers, who were able to use vapor deposition to generate thin samples of organic, atomic, and polymer glasses with exceptional kinetic, thermodynamic, and mechanical stability[7–13]. Key to the success of this technique is the idea to use the temperature of the substrate on which the material is deposited as a tuning parameter, which has to be optimized to maximize the ratio of the surface to bulk mobility.

Despite its attractivity, this approach does have some limitations[12]. Firstly, producing ultrastable bulk samples via vapor deposition is challenging because a slow deposition rate, of the order of 100 microns/day, is required. Secondly, the produced samples are sometimes markedly anisotropic due to the layer-by-layer deposition process on the free surface[12]. Thirdly, the technique cannot be applied easily to colloidal glasses, because, for these materials, it is not easy to deposit the particles at a different temperature from that of the substrate, see e.g., refs. 14,15.

Generating a well-annealed glass is a challenging task for computer simulations too[16], a difficulty that is shared with optimization problems in computer science[17]. Although many sophisticated algorithms have been proposed to tackle this problem, such as the replica-exchange Monte-Carlo (MC) method or the shoving algorithm[18–20], these methods either require a CPU time that depends super-linearly on the number of particles, or are too system specific. Recently, a particle-swap MC algorithm[21,22] was strongly optimized[23], opening a new way to prepare in silico ultrastable glasses with even higher stability than experimental samples[24]. Unfortunately, however, the particle swap moves used by these algorithms are extremely hard to realize in experiments.

An alternative approach to study the properties of ultrastable glasses is the so-called random pinning[25,26]. After having equilibrated the liquid, one freezes permanently the positions of randomly selected particles. It can be shown that this method allows to access real equilibrium states of the pinned system even in the ideal glass state[27–29], and it can be experimentally realized in colloidal[30,31] and molecular[32,33] systems. However, while random pinning gives important insight into the thermodynamic behavior of a bulk ideal glass[26–29], it also breaks translational invariance and, as a consequence, vibrational motion[34,35],

[1]Laboratoire de Physique de l'Ecole normale supérieure, ENS, Université PSL, CNRS, Sorbonne Université, Université Paris-Diderot, Sorbonne Paris Cité, Paris, France. [2]Department of Physics, Kyoto Sangyo University, Kyoto, Japan. [3]Laboratoire Charles Coulomb, University of Montpellier and CNRS, F-34095 Montpellier, France. ✉e-mail: f.zamponi@gmail.com

mechanical responses[36], and relaxation pathways[37–39] are radically altered, thus preventing one to gain insight into the dynamical properties of ultrastable glasses in bulk.

All these approaches share the underlying idea of modifying the behavior of certain degrees of freedom (e.g., surface mobility in vapor deposition, particle sizes in swap MC, pinned particles' movement in random pinning) by freeing or freezing them, thus allowing the glass-former to equilibrate quickly when the degrees of freedom are freed and obtain enhanced stability of the system when they are subsequently frozen[40,41]. In other words, this amounts to tune the height of the barriers in the energy landscape of the system by introducing or removing constraints on some degrees of freedom (Fig. 1). Note that if one wants to produce equilibrium configurations of the frozen system, it is extremely important to perform a quiet freezing[42] of these selected degrees of freedom, i.e., a process that preserves the equilibrium measure[26–28,43], or else the glass will be affected by out-of-equilibrium effects.

In this work, we overcome several of the problems mentioned above, by proposing a random bonding approach, in which one freezes the distance between a subset of neighboring particle pairs (Fig. 1). Random bonding is mathematically similar to random pinning, and we will show numerically that it satisfies the requirement of quiet freezing, hence producing configurations of the bonded system that are close to equilibrium, yet without breaking the translational invariance contrarily to random pinning. Hence, this approach allows to generate equilibrium glass configurations of a bulk system of molecules (e.g., dimers, trimers, and polymers) with widely tunable stability. Most importantly, this method can be implemented in real experimental systems such as colloids with patches[44–47] or DNA-linkers[48–50] and polymers[51,52], i.e., systems for which the interactions between the constituent particles can be modified via external parameters, see below for details.

Here, after having introduced this novel method, we present results from computer simulations that demonstrate the effectiveness of random bonding by determining for a simple glass-former the kinetic stability via heating-cooling cycles and mechanical stability via stress-strain measurements. Furthermore, we show that the generated configurations are close to equilibrium, such that no aging is detected within numerical precision. Finally, we discuss the experimental feasibility of the method.

# Results

## Protocol

Our protocol to generate ultrastable glass configurations of the bonded system goes as follows (Fig. 1 and Methods):

1. We first prepare, by standard cooling from high temperature, an equilibrium configuration of the non-bonded (monomer) system at the target temperature $T$.

2. Subsequently, we pick at random two particles that are nearest neighbors, and permanently fix their distance, thus creating a dimer. We repeat this process, thus creating a mixture of monomers and dimers that forms the ultrastable glass. The process stops when we reach the target dimer concentration $c$.

3. This mixture can now be studied via molecular dynamics (MD) simulations to study its thermal and mechanical response. The target values of temperature $T$ and dimer concentration $c$ can be tuned to obtain a glass with the desired stability.

Before proceeding to characterize our glasses, we stress once more that the non-bonded system serves only as an auxiliary preparation tool (Fig. 1). The ultrastable glass we are investigating in the following is created via the random bonding process (which, in our scheme, would be akin to a vapor deposition or random pinning) with target temperature $T = 0.42$ (at which the non-bonded system has already a sluggish dynamics[53]) and dimer concentration $c = 0.95$ (thus keeping 5% of monomers). Subsequently, we will discuss a phase diagram of stability in the $(T, c)$ plane.

## Kinetic stability

The kinetic stability of a glass-former is usually determined by a heating-cooling cyclic process[8,54,55]. We start from the configuration created by our random bonding protocol at $T = 0.42$, and then melt the glass by heating the system (keeping the bonds frozen) up to a high temperature, $T = 2.1$, using a constant rate $K = |\frac{dT}{dt}| \simeq 6 \times 10^{-3}$. Subsequently, we cool the system down to very low $T$ ($T = 0.001$), and we then heat it up a second time, using the same cooling/heating rate, $K \simeq 6 \times 10^{-3}$. Figure 2 shows the temperature evolution of the potential energy per particle and of the specific heat of the system. We find a very pronounced hysteresis, characteristic of ultrastable glasses[8,54]. During the first heating, our ultrastable glass remains in the glass state up to its kinetic glass melting (or devitrification) temperature $T_m^{usg} \approx 1.4$. When the system is subsequently cooled down, it remains in the liquid state down to its normal glass transition temperature, which

(a)

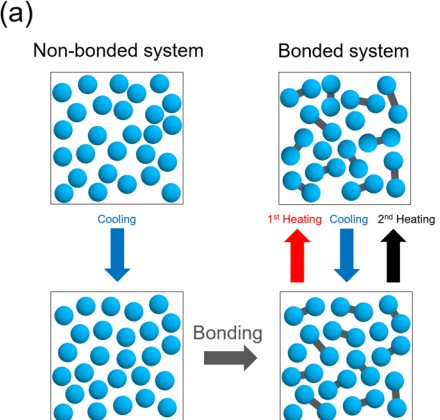

(b)

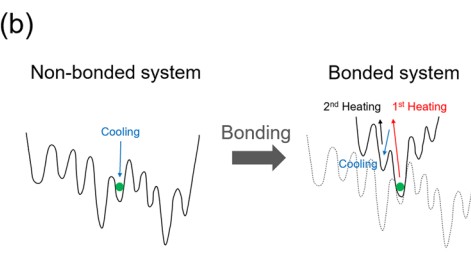

**Fig. 1 | Schematics of the protocol. a** We begin by creating an equilibrium configuration of the non-bonded (monomer) system by standard cooling to a target temperature $T$ at which the non-bonded system is still not very glassy. Note that a non-bonded system is only an auxiliary tool in our construction, while our aim is to construct an ultrastable glass of the bonded system. This is achieved by introducing the bonds via random bonding at the target temperature $T$. We now have a configuration of the bonded system, which is an ultrastable glass. To prove this, we perform heating/cooling cycles to measure kinetic stability. **b** Sketch of the (free) energy landscape. The non-bonded system is studied at temperatures such that barriers can be easily crossed. Once bonds are introduced, barriers become higher and the system remains trapped in a low-energy ultrastable glass basin. The first heating will allow the system to access high energy states, but subsequent cooling/ heating will only permit to reach configurations that have significantly higher energy than one of the ultrastable glass configurations.

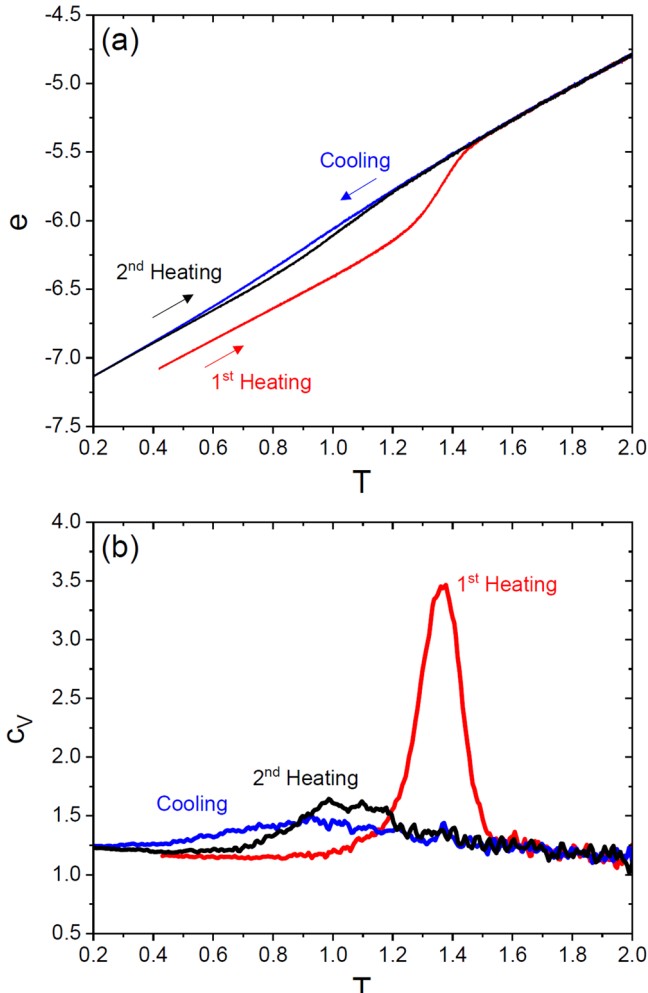

**Fig. 2 | Heating-cooling process. a** The potential energy per particle $e$ during the heating-cooling cyclic process with $K \simeq 6 \times 10^{-3}$. The curves are obtained by averaging over five different samples. The standard deviation due to sample-to-sample fluctuations is around 0.01 at $T \approx 1.35$, which is negligible in this plot. The system size is $N = 32400$. **b** Specific heat $c_V$ obtained from $c_V = de/dT$ using the data in (**a**). The standard deviation due to sample-to-sample fluctuations is around 0.1 at the maximum of the specific heat.

is estimated to be around $T_g \approx 0.8$, a value obtained from the $T$-dependence of the specific heat, Fig. 2b. Note that this simple cooling is not able to reach an ultrastable glass state, as shown by the higher value of the energy (Fig. 2a), and the hysteresis cycle obtained by re-heating the sample with the same heating rate (the second heating) is much smaller, indicating that the glass produced via a standard cooling procedure is indeed kinetically much less stable than the original ultrastable glass. Figure 2b shows the corresponding specific heat $c_V$, which makes the hysteresis even more visible. We have checked that similar results are obtained using a smaller rate $K$ (see Supplementary Fig. S6). We thus conclude that our bonding protocol has indeed created a glass-former with extremely high kinetic stability, compared to what can be achieved by simple cooling of the bonded liquid from high temperatures.

**Mechanical stability**

We examine the mechanical stability of our bonded systems by following the strategy of ref. 56. We first cool down our ultrastable glass, prepared at $T = 0.42$ and $c = 0.95$ as described above, to $T = 0.001$ at a rate $K \simeq 6 \times 10^{-3}$, and then quench it further to zero temperature by using an energy minimization algorithm. Subsequently, we perform a

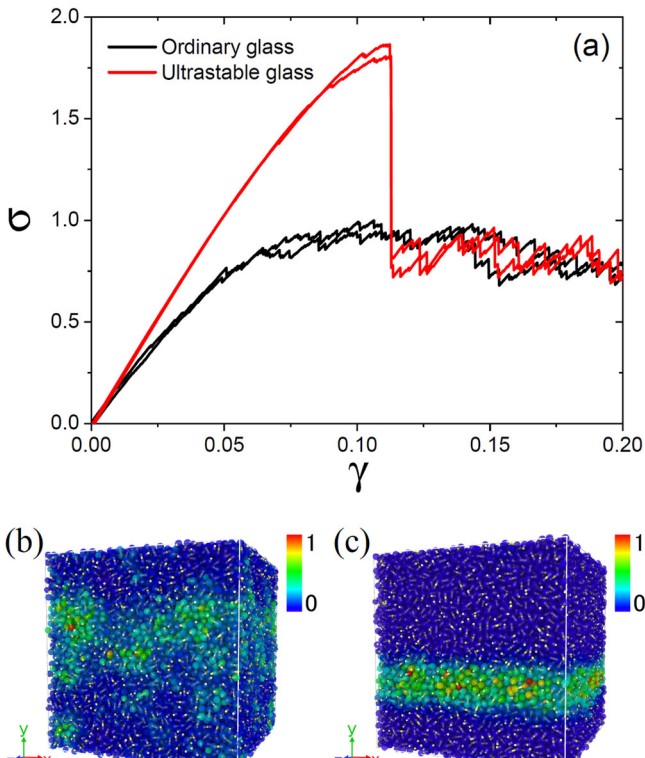

**Fig. 3 | Mechanical test. a** Stress versus strain curves of bonded glasses for ordinary and ultrastable samples. Two independent realizations are shown. Note that athermal quasi-static simulations are fully deterministic, hence no statistical errors are present. Also shown are snapshots of typical samples for an ordinary glass at $\gamma = 0.15$ (**b**) and an ultrastable glass at $\gamma = 0.12$ (**c**), respectively. The color bar corresponds to the non-affine displacement, $D_{min}^2$, measured from the origin, $\gamma = 0$[70]. The bonds between particles are shown in yellow.

standard zero temperature quasi-static shear simulation using Lees-Edwards boundary conditions. For comparison, we also consider a normally-annealed (ordinary) glass sample of the same (bonded) system, by using the configuration obtained after the first heating and cooling cycle shown in Fig. 2 (also quenched to zero temperature by energy minimization) as a starting point for the same quasi-static shear simulation.

In Fig. 3a, we show the shear stress $\sigma$ as a function of strain $\gamma$ for the ultrastable and the ordinary glass samples. The latter displays a mild stress overshoot and a gentle decrease of the stress after the maximum, indicating that the sample is ductile, at least for the system size $N = 32,400$ (see refs. 57–59 for a discussion of finite size effects). A real-space snapshot of the non-affine displacement field after the overshoot is presented in Fig. 3b, and shows a mild localization of shear. Remarkably, for the ultrastable sample, the stress overshoot is significantly enhanced, and an abrupt discontinuous stress drop emerges. This brittle yielding is accompanied by a sharp system-spanning shear band, as evidenced in Fig. 3c. The enhanced brittleness and mechanical stability observed in Fig. 3 confirms that the system is located in a very deep local minimum of the rugged energy landscape of the bonded system, due to the preparation protocol we used, and it thus corresponds to an ultrastable glass of the bonded system.

We also note that in ref. 36 the mechanical yielding of a randomly pinned glass was studied and no brittle yielding with a system-spanning shear band was observed, likely because the pinned particles break translational invariance. This demonstrates that our random bonding procedure has a huge advantage over random pinning, in that

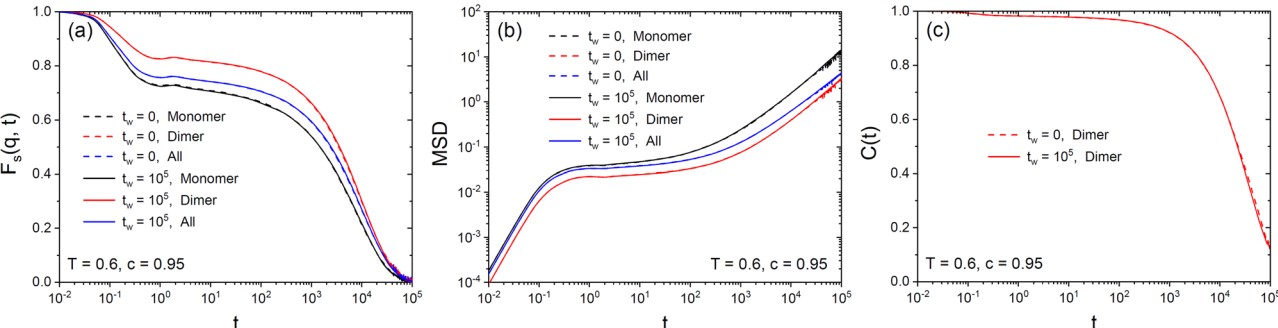

**Fig. 4 | Absence of aging after random bonding. a** Self-intermediate scattering function $F_s(q, t)$ for $T = 0.6$ and $c = 0.95$ for monomers, dimers and all particles. Dashed and solid curves indicate $F_s(q, t)$ computed from trajectories with the waiting time after the creation of the bonds $t_w = 0$ and $t_w = 10^5$, respectively. **b** The corresponding mean-squared displacement. **c** The corresponding rotational correlation function $C(t)$ for the dimer molecules. In all panels, statistical errors due to thermal fluctuations for a given sample are smaller than the line width. Data were then averaged over ten independent samples, and sample-to-sample fluctuations give a relative statistical error on the mean of about 5%, which is not reported in the plots for clarity. Note that such fluctuations are related to dynamical heterogeneity, and will be studied more systematically elsewhere.

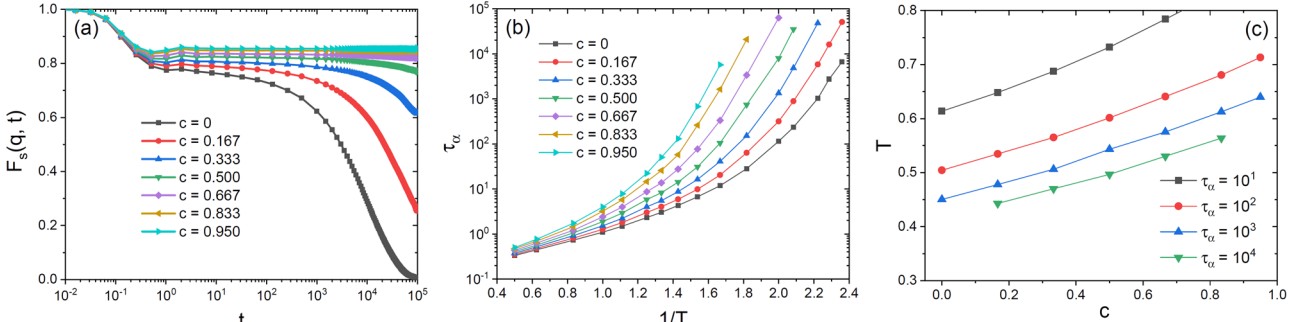

**Fig. 5 | Slow dynamics by random bonding. a** Self intermediate scattering function at $T = 0.4238$ and different concentrations $c$. The system size is $N = 1200$. **b** Structural relaxation time, $\tau_\alpha$, as a function of the inverse temperature, $1/T$, for several fixed concentrations $c$. **c** Lines of constant $\tau_\alpha$ in the $(T, c)$ plane, deduced from the data in panel b. In all panels, relative statistical errors are of the order of 5%, as discussed in Fig. 4.

it allows to investigate the mechanical properties of realistic bulk stable glasses.

## Absence of aging after random bonding

We now show that random bonding has the same quiet freezing property as random pinning, i.e., the system is already in thermal equilibrium right after the construction of the bonds. This feature thus allows to study the equilibrium dynamics, even in very deep glass states, simply by running a normal MD simulation starting from the configuration obtained right after bonding. Since, in the present case, this property holds only approximately, a more complete discussion of its validity will be given elsewhere.

To check equilibration, we chose a target temperature of $T = 0.6$ and a target dimer concentration of $c = 0.95$, such that the dynamics of the bonded system is slow, but not fully arrested on the simulation timescale. In Fig. 4a, we show the self-part of the intermediate scattering function, $F_s(q, t)$, defined for monomers, dimers, and all particles (see Supplementary Note 2 for the exact definitions). The wave number $q$ is 7.25, near the location of the main peak in the static structure factor. We observe full relaxation within the time window considered, and most importantly, that the correlation function measured right after the system is prepared ($t_w = 0$) and after a starting time $t_w = 10^5$ are numerically indistinguishable, which confirms that the bonding protocols produce equilibrated configurations, at least within our numerical precision. We confirmed the same behavior for other observables, and the mean-squared displacement and the rotational correlation function for the dimers (see Supplementary Note 2 for

details) are presented in Fig. 4b, c, respectively, consolidating the absence of aging.

## Equilibrium dynamics

We conclude our study by investigating systematically the equilibrium dynamics in the $(T, c)$ plane. In Fig. 5a, we show $F_s(q,t) = F_s^{All}(q,t)$ measured for all particles at $q = 7.25$, after preparation of the bonded system at the target temperature $T = 0.4238$ for different concentrations $c$. (See Supplementary Fig. S4 for corresponding results for other values of $T$.) As argued above, this corresponds to the equilibrium dynamics of the bonded system. For the original KA model, $c = 0$, we observe full relaxation within the time window considered, although at this $T$ the dynamics is already glassy. The dynamics slows down very quickly as $c$ is increased and for $c > 0.5$, it is completely frozen on the timescale of the simulation, demonstrating that the system has entered a truly glassy regime. We define the structural relaxation time $\tau_\alpha$ via $F_s(q, \tau_\alpha) = 1/e$ and in Fig. 5b, we show its temperature and $c$-dependence in an Arrhenius plot. From this graph, one concludes that increasing $c$ leads to a quick slowing down of the relaxation dynamics, in agreement with the result shown in Fig. 5a, and that this effect is strongly enhanced if $T$ is decreased, i.e., a behavior that is qualitatively similar to the one found for randomly pinned systems[27,28]. From these results, we can draw iso-$\tau_\alpha$ lines in the $(T, c)$ plane that characterize the stability of the glasses prepared by random bonding at these values of target temperature and dimer concentration (Fig. 5c).

Based on an extrapolation of these results, we estimate that the glass analyzed in Figs. 2, 3, prepared at $T = 0.42$ and $c = 0.95$, corresponds to an equilibrium relaxation time $\tau_\alpha \approx 10^{12}$, thus about a factor

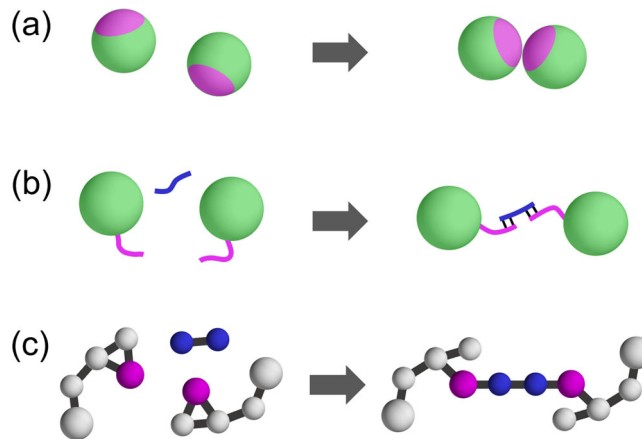

**Fig. 6 | Examples of bond formation processes in experimental systems.**
**a** Inducing attractive interactions between patchy colloids by changing the salt concentration. **b** Varying temperature or irradiation with ultra-violet light in colloids with DNA-linkers. **c** Chemical bonding between epoxy groups with a curing agent.

of $10^7$ larger than the largest $\tau_\alpha$ we accessed in our simulations. This demonstrates that the bonding process does indeed allow to generate glasses with a fictive temperature that is far lower than one can access by means of standard algorithms, and comparable to that achieved by the swap algorithm[23].

Finally, it has been reported that, in contrast to conventional glass-forming liquids, the equilibrium dynamics of pinned systems show a decoupling between self and collective relaxation and suppression of dynamical heterogeneity upon approaching the glass transition, due to the highly confined environment[37,39]. We report that the randomly-bonded systems do not show such a decoupling between the self and collective parts (see Supplementary Fig. S5), as expected for a system that has translational invariance. We leave the study of the dynamical heterogeneity of randomly-bonded systems for future work.

## Discussion

We have introduced a novel protocol that allows one to prepare highly stable equilibrium glasses in bulk (Fig. 1). The approach requires the generation of an equilibrated configuration of a simple glass-forming system at intermediate temperatures, i.e., where its dynamics is not yet arrested (which is routinely done in experiments and in simulations), and subsequently the introduction of random bonds between neighboring molecules (here monomers) to form a certain fraction of dimers. We numerically demonstrated that the resulting configurations of the bonded system are in equilibrium, and our simulations show that their dynamics is extremely slow, indicating that one does indeed access deep glassy states. Moreover, we have demonstrated the strong enhancement of kinetic and mechanical stability via a non-equilibrium heating-cooling process and athermal quasi-static shear simulations, respectively.

We emphasize that it is possible to realize the proposed random bonding in real experiments of soft matter systems, such as colloids and emulsions, for which existing vapor deposition techniques cannot be applied easily, thus allowing to produce ultrastable glasses for a new class of glass-formers. One can, e.g., tune interactions between colloidal particles, by introducing patches or DNA-linkers[47,48], as schematically shown in Fig. 6a, b. Importantly, one can turn on the bond interactions at any time by varying the concentration of salt or the temperature[48,50], or using ultra-violet light[49]. To demonstrate this, we also carried out experiments using two-dimensional patchy colloids. Starting from an equilibrated binary mixture of patchy and non-patchy colloids (monomers)[60], we increase the salt concentration to activate

the patchy interaction between patchy colloid pairs, leading to the formation of dimers with strong bonds. The resulting configurations have a finite concentration of dimers (with a few trimers and tetramers), demonstrating that, indeed, one can turn on the bond interactions at any time (see Supplementary Note 3 for details). A further possibility to implement random bonding in molecular or polymeric systems is schematically shown in Fig. 6c. For epoxy resins, e.g., one can control polymerization by adding to the sample a curing agent[51,52] or exposing it to ultra-violet light[61]. These few examples illustrate that it is in principle, possible to produce ultrastable glasses by particle bonding in experiments, thus opening new research directions in the field.

The random bonding between two neighboring particles to produce dimer systems is, of course, only one possible implementation of our approach, and it is straightforward to extend the method to construct trimers, polymers, or more complex molecules. We expect that more stable configurations can be obtained when a particle can have more bonds, due to the freezing of additional degrees of freedom[41,62–64]. When the number of bonds increases, a network structure percolates through the system, leading to a glass-gel crossover[60,65,66]. We expect that the random bonding will thus provide a recipe to produce ultrastable gels as well, thus allowing us to study also the glass-gel crossover in equilibrium.

By exploring these ideas, it should be possible to realize an ideal glass transition in a bulk equilibrium bonded system akin to the randomly pinned systems[26–28], yet without breaking translational invariance. Hence we expect that our study will pave the way to an experimental realization of ideal bulk glasses.

## Methods

As a monomer system, we use the standard Kob-Andersen (KA) model in three dimensions[67]. The original, non-bonded (monomer) KA model is simulated in the $NVT$ ensemble with $N$ particles in a periodic volume $V$ at the target temperature $T$ (see Supplementary Note 2) to produce the initial equilibrium configuration. We then introduce a certain fraction of diatomic dumbbell molecules, i.e., dimers, by random bonding particles that have a distance less than $R_b = 1.5$. (All quantities are expressed in standard reduced Lennard-Jones units[67].) In this way, we create a system with $N_m$ monomers and $N_d$ dimers. By construction, $N = N_m + 2N_d$. The dimer concentration is $c = \frac{2N_d}{N} = \frac{N - N_m}{N}$. To simulate the bonded system at finite temperature, we use the RATTLE algorithm to keep the bond lengths fixed[68]. To study mechanical responses, we use an athermal quasi-static shear simulation via an energy minimization algorithm[69], in which the rigid bonds were replaced by a harmonic spring with sufficiently hard stiffness, see Supplementary Note 2 for details.

## Data availability

The data necessary to reproduce the figures are publicly available through the Zenodo Repository at https://doi.org/10.5281/zenodo.7435970.

## Code availability

Simulations for the non-bonded system (KA model) were carried out using the LAMMPS package (www.lammps.org). Scripts are available from W.K. (walter.kob@umontpellier.fr) upon reasonable request. Simulation codes for the bonded systems are available from M.O. (misaki.ozawa@univ-grenoble-alpes.fr) upon reasonable request.

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

## Acknowledgements
We thank J.-L. Barrat, M. Ediger, V. F. Hagh, H. Ikeda, F. C. Mocanu, M. A. Ramos, F. Sciortino and P. G. Wolynes for discussions, and K. Yoshihara for his contribution to the preliminary patchy particle experiment. This project has received funding from the European Research Council (ERC) under the European Union's Horizon 2020 research and innovation program (grant agreement n. 723955 - GlassUniversality). W.K. is member of the Insitiut universitaire de France.

## Author contributions
M.O., Y.I., W.K., and F.Z. designed the research. M.O. and W.K. performed the computational research. Y.I. designed and performed the experimental research. M.O., Y.I., W.K., and F.Z. analyzed the data and wrote the paper.

## Competing interests
The authors declare no competing interests.
