## [Peer Review File · Nature Communications]

Creating bulk ultrastable glasses by random particle bondingREVIEWER COMMENTS

Reviewer #2 (Remarks to the Author):

In this paper, the authors present a novel computational approach, founded on random particle bonding, to generate ultrastable glasses, with enhanced thermal and mechanical stability, as in the experimental counterpart. According to the authors, this approach can be translated into experimental physics, yielding ultrastable bulk glasses. This would signify a novel approach since, up to this moment, only vapour deposited thin film ultrastable glasses have been produced.

In my opinion, one of the main claims and interests of the presented work is the possibility to extrapolate the simulation analysis to the experimental world and, hence, obtaining bulk ultrastable glasses. Unfortunately, the explanation given by the authors concerning this point is not convincing to me. The authors give some examples of preparation of ultrastable glasses following the approach undertaken during their computational research. For example, tuning the bond interaction between molecules varying salt concentration, temperature or by exposure to ultraviolet light (controlling polymerization of chains). These experiments, as far as I understand, would not yield an ultrastable glass. Let me explain why.

For vapour deposited ultrastable glasses, a particular material is evaporated and deposited onto a substrate. Thanks to the enhanced surface mobility or mediated by a particular relaxation process active on the surface, the system can reach equilibrium faster than in the bulk. So at a temperature where bulk mobility is so large that the system cannot be equilibrated, we can an equilibrated glass thanks to this surface-mediated pathway, forming an ultrastable glass. Now, when one heats this ultrastable glass above the glass transition temperature for a given (large) amount of time, the system equilibrates back to the same equilibrium state that would reach heating an ordinary liquid-quenched glass, with the same structure at a molecular level. The different between an ordinary and ultrastable glass is, hence, only the packing density and the molecular arrangement. However, if one takes a molecular system and induced crosslinking or any physical process that may alter the structure of the system, the initial structure is no longer recovered once heated above T_g . Even in the liquid state, the system is now different because of this chemical or physical modification of the network of the material. Of course, by increasing the degree of cross linking, or the molecular weight of the polymer (forming dimers or trimers), the devitrification temperature of the system increases and the mechanical properties changes, but due to a reason fundamentally different to that of ultrastability. This is interesting and important by itself but considering that ultrastable glasses are the main point of the manuscript, I think this point should be strongly considered.

This raised issue concerns the extrapolation that the authors do towards the experimental realm and do not pose questions regarding the computational approach to generate ultrastable glasses. Indeed, as nicely shown in Figure 2, the heating/cooling curves indicate how increasing bonding concentration yield a more stable glass, with higher devitrification temperature, while keeping the same glass transition temperature on cooling (as far as I see, because this, by the way, should be indicated more clearly). In the case of the experimental counterpart, I would expect that, upon, for example, crosslinking, the devitrification temperature would indeed increase but, at the same time, the glass transition on cooling will also increase, keeping, a priory, the same "hysteresis level" unperturbed.

Unfortunately, at this stage, I am not convinced that the degree of novelty and impact of this work justifies the publication in Nature Communications. What does this work provide, in comparison to the other computation methods to generate ultrastable glasses, such as the Berthier's particle swap, apart to the unjustified (in my opinion) possibility to extrapolate it towards the experimental side?

Considering the quality of the work and the new material it provides to the field of glasses and ultrastable glasses, I would gladly accept to review a new version as long as the authors are able to provide more solid proofs concerning the applicability of this method to produce ultrastable glasses, or deeply reformulate the goal of the paper.

Reviewer #3 (Remarks to the Author):

The manuscript by Ozawa et al. describes a methodology that allows creating highly stable glassy configurations by randomly imposing the formation of directional bonds between pairs of particles within a disordered ensemble. They test the stability of the computational glass by heating/cooling cycles where they observe an increase of the onset temperature of devitrification in close agreement to what is reported with experimental ultrastable glasses. They further show the impact of the increased stability in the mechanical response of the glass by performing stress-strain test. In such plots the stable glass shows an abrupt discontinuity indicative of the formation of well-defined shear bands.

I find the manuscript very interesting and relevant, but I have doubts about its suitability for Nature Communications. Before I could recommend publication I would like the authors to address the following comments:

1.- What are the main differences of the proposed methodology to the Kern-Frenkel model for patchy colloids [Kern, N. & Frenkel, D. J. Chem. Phys. 118, 9882–9889 (2003)] and subsequent modifications? see for instance, Smallegange and Sciortino, Nature Physics, 9, 554–558 (2013). This approach allows to stabilize liquids down to very low temperatures.

2.- It is repeated several times throughout the text the impossibility of vapor depositing colloidal particles to mimic the process employed to create atomic or molecular glasses. This is not strictly true, there are techniques that allow for deposition of disordered or crystalline arrays of colloidal small particles. See for instance: Nguyen and Choi, Scientific Reports, 10, 11075 (2020). The fact that has not been done does not mean that cannot be done.

3.- The manuscript is accompanied by a final section that shows it is indeed possible to experimentally prepare colloidal glasses with well-defined interactions between particles. There are already quite a few reports in the literature dealing with the formation of structures by using patchy particles and directional interactions, so it would be useful for the non-specialized reader to contextualize the present approach with previous work.

I highly appreciate the effort of experimentally showing the formation of such a glass but in my opinion the manuscript would be much stronger if it could present some evidence that these patchy colloidal glasses are indeed located in lower energy configurations. If it is not possible, I do not see the point of adding this section since patchy colloidal glasses have been created previously.

4.- The kinetic and mechanical test are suitable ways to demonstrate the creation of

computer ultrastable glasses. I hope in future works the authors can address more specifically the glass transition of these materials and whether it occurs by a front mediated mechanism or by phase separation as observed in previous works in both computational and experimental ultrastable glasses.

Reviewer #2

We thank the referee for the positive comment, “This would signify a novel approach since, up to this moment, only vapour deposited thin film ultrastable glasses have been produced.”, as well as for their constructive criticism. Below we describe our responses to the referee’s comments and the changes made accordingly.

Referee’s comment:

For vapour deposited ultrastable glasses, a particular material is evaporated and deposited onto a substrate. Thanks to the enhanced surface mobility or mediated by a particular relaxation process active on the surface, the system can reach equilibrium faster than in the bulk. So at a temperature where bulk mobility is so large that the system cannot be equilibrated, we can an equilibrated glass thanks to this surface-mediated pathway, forming an ultrastable glass. Now, when one heats this ultrastable glass above the glass transition temperature for a given (large) amount of time, the system equilibrates back to the same equilibrium state that would reach heating an ordinary liquid-quenched glass, with the same structure at a molecular level. The different between an ordinary and ultrastable glass is, hence, only the packing density and the molecular arrangement. However, if one takes a molecular system and induced crosslinking or any physical process that may alter the structure of the system, the initial structure is no longer recovered once heated above T_g . Even in the liquid state, the system is now different because of this chemical or physical modification of the network of the material. Of course, by increasing the degree of cross linking, or the molecular weight of the polymer (forming dimers or trimers), the devitrification temperature of the system increases and the mechanical properties changes, but due to a reason fundamentally different to that of ultrastability. This is interesting and important by itself but considering that ultrastable glasses are the main point of the manuscript, I think this point should be strongly considered.

This raised issue concerns the extrapolation that the authors do towards the experimental realm and do not pose questions regarding the computational approach to generate ultrastable glasses. Indeed, as nicely shown in Figure 2, the heating/cooling curves indicate how increasing bonding concentration yield a more stable glass, with higher devitrification temperature, while keeping the same glass transition temperature on cooling (as far as I see, because this, by the way, should be indicated more clearly). In the case of the experimental counterpart, I would expect that, upon, for example, crosslinking,

the devitrification temperature would indeed increase but, at the same time, the glass transition on cooling will also increase, keeping, a priori, the same “hysteresis level” unperturbed.

Our response:

The referee’s main criticism is that if one alters the system by bonding, the devitrification temperature increases since the system changes, which leads to an *apparent* ultrastability compared with the system *without* bonding (monomer system). We totally agree with this conclusion. This remark of the referee made us realize that our description of the protocol regarding the production of the ultrastable glass was not sufficiently clear and therefore needed to be improved. Our protocol aims in fact to produce bonded glasses having ultrastability compared with *the same* bonded system. To explain our core idea more clearly and to avoid misunderstanding, we completely reformulated the main text, as the referee suggested. In Fig. 1 of the revised manuscript, we show a new schematic figure explaining the concept of our protocol. First, we prepare an original non-bonded (monomer) system at a low temperature using a standard cooling procedure. Of course, this naive slow cooling does not produce ultrastable glasses as all previous methods failed to do so (except swap Monte-Carlo simulations). Yet this process is crucial, as we will explain below. We then perform random bonding to create our ultrastable glass sample. To assess ultrastability of this sample, we perform the heating-cooling cycle. The 1st heating process involves a huge energy jump (see Fig. 2(a) in the revised manuscript) associated with a large peak in the specific heat c_V (see Fig. 2(b), new). At a high temperature, this bonded glass melts. We then cool the sample again, which corresponds to a poorly-annealed or ordinary glass sample *with the same chemical composition*. We find that during the 2nd heating process this sample shows a mild temperature dependence with a small c_V peak, as expected for a glass that has been prepared via a normal cooling procedure. Therefore, the initial bonded glass demonstrates a huge stability during the 1st heating compared with the same bonded system during the 2nd heating after one heating-cooling cycle. Thus, this procedure directly demonstrates ultrastability of the initial bonded glass sample compared with the same bonded glass after standard cooling. We stress once again that the chemical composition of the system is never altered during these heating/cooling cycles.

To understand the origin of ultrastability of our protocol more conceptually, we have now added the new Fig. 1(b) to the main text, in which we present the energy landscapes in the phase space. The cooling for the original non-bonded (monomer)

system gives rise to a state point (shown in green) inside a (not very deep) local energy minimum. The random bonding introduces constraints in the phase space by freezing degrees of freedom associated with inter-particle distances. This corresponds to confining the energy landscape, and now the initial green state point is deep inside the local minimum of the new energy landscape associated with the bonded system. Once the system is heated and the state point goes out from the minimum, it is very hard to come back to such a deep minimum by standard cooling of the bonded system.

To provide a more intuitive explanation, we take an analogy with the Tetris game (as often mentioned in Ediger’s literature and presentations). In Fig. 1 of the present reply, we highlight the difference between the vapor deposition method (left) and our protocol (right). In the vapor deposition case, a slower deposition rate is required to prepare well-packed, ultrastable glasses, otherwise poorly-packed glasses are obtained at a faster rate. This is essentially the same for the standard cooling approach in the sense that the system needs a longer time to achieve a well-packed structure. Our protocol is totally different. First, we fill the container with the single boxes (monomers), which corresponds to the 1st cooling of the original non-bonded system. We then make bonds randomly, creating the Tetris blocks (molecules). Obviously, by construction, the new bonded configuration is very well-packed. Using the language of the Tetris game, we do not solve the problem associated with many frustrations due to molecular shape. Instead, we create the problem such that the initial state is the optimized solution. This emphasizes the importance of the first cooling process for the original monomers, which plays a role in finding a packed structure that bonded molecules cannot attain.

Figure 1: An analogy with the Tetris game. Left: Cartoon explaining the ultrastability achieved by vapor deposition (or ordinary cooling) at a slower rate. The figure is taken from [*Physics Today* **69**, 1, 40 (2016); <https://doi.org/10.1063/PT.3.3052>]. Right: Cartoon showing our random bonding approach.

Therefore, the key experimental issue for our new protocol is to make bonding *after* preparing a supercooled liquid of monomers. This requires a technology to introduce or switch on bonding interactions *any time* in the supercooled liquid state. In the main text, we discuss patchy colloids varying salt concentration, DNA-linkers induced by ultra-violet light, and polymerization of molecules, as possible approaches. Besides, we demonstrate the feasibility in an experiment with patchy colloid by varying the salt concentration as a proof of concept of this technology (We now moved most of these results to SI following the suggestion of Referee #3). Clearly, our protocol is completely different from the standard studies, such as patchy colloids and polymers, where bonding interactions are present from the beginning.

To conclude, we believe that our protocol is truly innovative and in a totally different paradigm from the vapor deposition and ordinary cooling methods, but we agree with the referee that in the original manuscript this was not explained properly. In particular we now make it clear that we compare the properties of the ultrastable glass with the glass-former that has the *same* composition, thus the comparison is fair. We hope that the revised manuscript will do a better job in explaining what we actually did.

Reviewer #3

We thank the referee for the critical evaluation of our work and for describing it as “very interesting and relevant”.

Referee’s comment:

1.- *What are the main differences of the proposed methodology to the Kern-Frenkel model for patchy colloids [Kern, N. and Frenkel, D. J. Chem. Phys. 118, 9882–9889 (2003)] and subsequent modifications? see for instance, Smalenburg and Sciortino, Nature Physics, 9, 554–558 (2013). This approach allows to stabilize liquids down to very low temperatures.*

Our response:

To the best of our knowledge, all previous literature, including the Kern-Frenkel model and its variants, studied the situation where bonded interactions between two particles are present from the beginning. On the contrary, our study switches on the bonding *after* making densely packed monomers. This is the crucial point to achieve ultrastability, as we described in response to Referee #2. If one prepares a system with bond interactions from the beginning, as previous literature studied, the standard cooling results in an ordinary, poorly-annelaed, glass sample. In order to clarify this important point we have now added the (new) Fig. 1 in which we illustrate our procedure. Also Fig. 2 has been strongly modified so that it becomes clearer that there is a huge difference between our ultrastable glass sample and a glass sample *of the same system* that has been produced by the usual cooling technique.

Regarding the paper by Smalenburg and Sciortino, the meaning of stability mentioned in their paper is totally different from the one we used in our manuscript. In their case, stability refers to the numerical observation that the free energy of a liquid state at a very low temperature is lower than the crystal ones (at least within known crystals). In this sense, the liquid phase is more stable than the crystal phase. This was achieved by specially tailored patchy colloid systems such that the system remains liquid down to zero temperature by tuning the patches, which controls the shape of the phase diagram. Instead, stability used in our study is concerned with the glass state, which depends on the preparation protocol. This notion of stability is related to non-equilibrium processes, such as heating-cooling cycles and mechanical deformation, which is totally different from the stability in the sense of equilibrium thermodynamics in the

paper by Smallenburg and Sciortino.

Nevertheless, one can employ previous patchy colloid simulation models, such as the Kern-Frenkel model, to realize our protocol in silico, instead of using rigid body inter-particle bonding as we did. The former would be more realistic when it comes to comparing with real experiments. However, we choose the rigid body constraint in this study because it allows us to do theoretical considerations and compare it with the random pinning method.

Referee's comment:

2.- It is repeated several times throughout the text the impossibility of vapor depositing colloidal particles to mimic the process employed to create atomic or molecular glasses. This is not strictly true, there are techniques that allow for deposition of disordered or crystalline arrays of colloidal small particles. See for instance: Nguyen and Choi, Scientific Reports, 10, 11075 (2020). The fact that has not been done does not mean that cannot be done.

Our response:

We thank the referee for pointing out to us the paper by Nguyen and Choi. Besides, we also found a paper studying the deposition growth of a colloidal glass [Cao, Zhang, and Han, Nat. Commun. **8**, 1 (2017)]. Nevertheless, both papers did not achieve ultrastable glass sample. In the revised manuscript, we cite these papers in the introduction part, in the sense that the deposition of colloids is doable but not easy and that it does not allow to produce ultrastable glasses.

Referee's comment:

3.- The manuscript is accompanied by a final section that shows it is indeed possible to experimentally prepare colloidal glasses with well-defined interactions between particles. There are already quite a few reports in the literature dealing with the formation of structures by using patchy particles and directional interactions, so it would be useful for the non-specialized reader to contextualize the present approach with previous work. I highly appreciate the effort of experimentally showing the formation of such a glass but in my opinion the manuscript would be much stronger if it could present some evidence that these patchy colloidal glasses are indeed located in lower energy configurations. If it is not possible, I do not see the point of adding this section since patchy colloidal glasses

have been created previously.

Our response:

We thank the referee for this comment, which is useful in improving the accessibility of the manuscript for non-specialized readers. In the revised manuscript, we now cite the following seminal papers studying patchy particles in the introduction part.

-Chen, Chul Bae and Granick, *Nature* **469**, 381 (2011)

-Chen, Yan, Zhang, Chul Bae and Granick, *Langmuir* **28**, 13555 (2012)

-Wang, Wang, Breed et al., *Nature* **491**, 51 (2012)

The big conceptual difference between our study and previous works is that the bonding between particles is switched on *after* making fairly densely packed (or super-cooled) monomers. As we already mentioned, this initially packed structure is a key point for our protocol to achieve ultrastability. To this end, the experimental challenge is to be able to tune the interactions between particles *at any time*, and in particular in a densely packed phase. This possibility is demonstrated by our preliminary patchy colloid experiment. Unfortunately, however, we were not able to push the technique as far as needed, and we decided to leave the assessment of the ultrastability of the sample for future study. Therefore in this revised manuscript, we opted to focus on proposing the main idea and its numerical verification and moved the experimental snapshot to the SI, as suggested by the referee.

Referee's comment:

4.- The kinetic and mechanical test are suitable ways to demonstrate the creation of computer ultrastable glasses. I hope in future works the authors can address more specifically the glass transition of these materials and whether it occurs by a front mediated mechanism or by phase separation as observed in previous works in both computational and experimental ultrastable glasses.

Our response:

We definitely agree with the referee that this is a very important point for future

studies, and we thank them for the suggestion.

REVIEWERS' COMMENTS

Reviewer #2 (Remarks to the Author):

The authors have fully address all of my previous concerns. As they explain, my concerns were due to a missinterpretation of their results that they have address in the revised version of the manuscript. Now, I can agree with their research, which, as I commented in the previous revisions, shows a new and original perspective on ultrastability which can stimulate their use for further fundamental and applied research.

For all this, I can now recommend publication of this manuscript in its present form in Nature Communications.